# Coordination measure for coupling system of digital economy and rural logistics: An evidence from China

Hui Shu[1], Lizhen Zhan[1]*, Xiaowei Lin[2], Xideng Zhou[3]

1 School of Business Administration, Jiangxi University of Finance and Economics, Nanchang, China,
2 School of Business, Minnan Normal University, Zhangzhou, China, 3 School of Economics and Management, Yuzhang Normal University, Nanchang, China

* zhanglz1124@163.com

**Data Availability Statement:** All relevant data are within the paper and its Supporting Information files.

**Funding:** (i) Hui Shu was funded by Jiangxi Social Science Key Fund Project: "Synergistic Research

## Abstract

As an important engine for high-quality economic development, the digital economy is gradually integrating with the rural logistics industry. This trend is contributing to making rural logistics a fundamental, strategic, and pioneering industry. However, some valuable topics remain unstudied, such as whether they are coupled and whether there is variability in the coupling system across the provinces. Therefore, this article takes system theory and coupling theory as the analytical framework, aiming to better elaborate the subject's logical relationship and operational structure of the coupled system, which is composed of a digital economy subsystem and a rural logistics subsystem. Furthermore, 21 provinces are seen as the research object in China, and the coupling coordination model is constructed, aiming to verify the coupling and coordination relationship between the two subsystems. The results suggest that two subsystems are coupled and coordinated in the same direction, and they feed back and influence each other. During the same period, four echelons are divided and there is variability in the coupling and coordination between the digital economy and rural logistics, according to the coupling degree (CD) and coupling coordination degree (CCD). Findings presented can serve as a useful reference for the evolutionary laws of the coupled system. The findings presented here can serve as a useful reference for the evolutionary laws of coupled systems. Moreover, it further provides ideas for the development between rural logistics and the digital economy.

## Introduction

The scale of the digital economy has been expanding, owing to the rapid development of a new round of digital technologies represented by artificial intelligence, blockchain, cloud computing, and big data (ABCD technology) [1]. As a result, with the participation of the digital economy, a new development standard has been established, and standardization, intelligence, and digitalization have gradually become the basic guidelines for continuous innovation in the fields of production, circulation, and consumption [2, 3]. Additionally, China's 14th Five-Year

on the Supply Chain Ecosystem of Smart Networked Vehicles in Jiangxi Province(22GL04)." (ii) Xiaowei Lin was funded by the National Social Science Foundation of China Western Project: "Research on the dilemma, causes, and strategies of high-quality development of rural logistics driven by digital economy"(No. 20XJY011). (iii) Xideng Zhou was funded by the Science and Technology Research Project of Jiangxi Provincial Education Department(No. GJJ213106). The funders had no role in study design, data collection and analysis, decision to publish, or preparation of the manuscript.

**Competing interests:** The authors have declared that no competing interests exist.

Plan (2021–2025) emphasized that smart logistics was a new growth point, which clarified the importance of the integration of the digital economy and the logistics industry. Benefitting from rural economic development, especially, electronic business prosperity, rural logistics has become a potential growth area of the logistics industry [4]. Rural logistics undertakes both the downward trend of industrial products and the upward movement of agricultural products and its development is related to the "money bags" of rural consumers and the "vegetable basket" of urban consumers. Hence, to develop a more effective logistics network in rural regions, the Chinese government has taken a series of measures to encourage logistics enterprises to establish a more efficient logistics network, which is a key component of the *Rural Revitalization Plan* [5]. However, the rural logistics industry has successively exposed some problems in its development. Due to the prevention and control of COVID-19, and the development of the new era, it is facing a series of difficulties, such as a weak industrial base, low penetration of digital technology, and weak ability to integrate resources. For example, the loss rate of agricultural products such as fruits and vegetables in China is four to five times higher in the logistics of picking, transportation, and storage [6]. As the digital economy and rural logistics gradually integrate, however, digital elements are gradually "penetrating" and "empowering" rural logistics, and digital transformation has become a new idea for the-quality development [7, 8]. Therefore, it is an important topic to research whether the digital economy and rural logistics have a coordinated development, which contributes to promoting a shift from quantitative growth to qualitative improvement in the rural economy.

As the digital economy and rural logistics keep more closely linked, the two are gradually coupling to become new drivers for economic development. With the incorporation of digital elements, digital technologies are being used in rural logistics. The use of drones and robots, for example, helps to visualize logistics activities, and ensures the availability of the products at the right place, time, and quantity and thus reducing wastage [9]. At the same time, digitalization enables the simulation of logistics activities and the optimization of tools. Such an environment enhances employee skills and innovativeness through the training programs, and reduces transportation emissions through optimum utilization [10].

The relationship between the digital economy and rural logistics has been the research hotspot, which focuses on three main aspects. Some scholars considered the importance of the digital economy for the development of the rural logistics industry. They focused on arguing that digital transformation was a necessary trend for high-quality development in the logistics industry. Brynjolfsson and Kahin (2003) [11] and Brynjolfsson and Collis (2019) [12] suggested that data was a new production factor that could be integrated into the rural logistics industry to provide better matching mechanisms and inspiring innovation in rural logistics. Sun et al. (2020) [13] and Yavas and Ozkan-Ozen (2020) [14] considered that digital logistics was an inevitable product of digital economy, and promoting the digital transformation of logistics was not only an important grasp to solve the bottleneck of development such as high cost and low efficiency, but also the main mode for digital transformation in rural logistics industry. Some scholars had studied how the digital economy and rural logistics were integrated. Existing research indicates that the digital economy is associated with rural logistics in two main ways. One is to add new impetus to rural logistics through the application of digital technology. For example, the introduction of technologies such as big data, cloud computing and blockchain in warehousing, transport, and distribution will upgrade traditional logistics into a modern digital logistics model. The second is that rural logistics rely on digital platforms for innovation and transformation, such as combining logistics big data platforms, logistics clouds, and smart logistics to promote changes in China's production, circulation and consumption methods [15]. Then, many scholars focus on how rural logistics could establish adaptive mechanisms to rationalize the influx of digital technologies and models. Pan et al.

(2020) [16] suggested that a large number of digital platform companies had entered the rural logistics market and became the backbone of the digital transformation of rural logistics. Then, Ma (2021) [17] and Wang et al. (2021) [18] also indicated that the demand for "iron and public infrastructure", "cloud and network" and digital logistics services would form a huge blue ocean market for rural logistics digitalization.

The term "coupling" is derived from physics and refers to the degree of dynamic interaction and influence between different systems [19]. When the sum of the assessed values is fixed, the closer the two assessed values are, the higher the degree of coupling (CD). However, the coupling model only characterizes the degree of interaction between systems and ignores the capacity level of each system [20, 21]. The coupling and coordination model, however, takes into account both coupling and coordination relationships and provides a more comprehensive measure of the interactions between systems [22]. For example, Zhang et al. (2019) [23] measured the degree of interaction between urbanized rural hollowing using the CCD model. Lai et al. (2020) [24] developed a comprehensive evaluation index system for ecology, economy, and tourism and calculated the coupling and coordination of the three subsystems in 31 Chinese provinces. Zhu et al. (2022) [25] established a CCD model and analyzed the dynamic characteristics and types of CCD between the two systems in Tangshan city. Lin et al. (2022) [26] used CCD model to reveal interactions between urban resilience subsystems. Xie et al. (2022) [27] studied the coupling relationship between cold-chain logistics and the economy from 2010 to 2019, based on the CCD model. Therefore, referring to the research of these scholars, we use the CCD model to measure the coupling and coordination between the digital economic system and the rural logistics system.

With the gradual penetration of digital elements into the real economy, digital transformation is gradually becoming a core element in the high-quality development of the rural logistics industry. Digital elements drive it to strengthen the digital foundation, make up for technical shortcomings, etc. However, current studies have mainly analyzed the importance, and impact of relationships and coordination paths between the digital economy and rural logistics. Few studies have examined their logical relationship as well as regional differences from a coupling perspective. Clarifying the coupling relationship becomes an important topic. Specifically, this study intends to answer the following questions.

1. Based on theoretical analysis, how the coupled system consisting of the digital economy subsystem and the rural logistics subsystem operates?

2. By constructing a coupling coordination model, whether it can be verified that the digital economy subsystem and the rural logistics subsystem are both coupled?

3. Is the evolution of the coupled system set in stone? If not, are there different stages of coupling coordination, and what kind of variability is there?

To answer these questions, this study sets up the evaluation index and constructs a coupling and coordination model to evaluate CD and CCD. On the one hand, system theory and coupling theory are used as the analytical framework to innovatively construct the coupling system, and clarify the main logical relationship and operation structure. On the other hand, based on the analysis results of CD and CCD, the changing trends and regional differences are examined. Ultimately, suggestions are provided for the development of the coupling between the digital economy and rural logistics.

The work of this paper is as follows: Section 2 sorts out the research on the relationship between digital economy and rural logistics from three aspects. Section 3 theoretically analyzes the components, logical structure, and operation level of the coupling system, which mainly consists of the digital economy subsystem and rural logistics subsystem. Section 4 establishes a

comprehensive evaluation index system to measure CD and CCD. Section 5 constructs a coupling coordination model to evaluate the coupling relationship. Section 6 evaluates the coupling relationships. Section 6 presents the innovations and limitations of this study based on the research context. Section 7 summarizes the three conclusions and discusses innovation and the relevance of the findings. Section 8 proposes adaptive management recommendations to inform the digital transformation of rural logistics.

## Literature review

The logistics industry is the front end of the application scenario, and it plays a critical role in rural economic development. Thanks to the innovative transformation of digital technology, rural logistics, and digital elements interact and interconnect, making rural logistics transform from a traditional value realizer to a value enhancer. Consequently, their coordinated development has become a hot topic of research for scholars.

The existing literature has studied the relationship between the digital economy and rural logistics from different perspectives, which can be summarised into three main themes by sorting them out, as follows.

Regarding the importance of the digital economy for rural logistics development, some scholars focused on arguing that digital transformation was an important direction for the transformation and development of rural logistics. With the acceleration of rural supply-side reform and the construction of a new rural digital economy, the key to agricultural development has shifted from the production field to the circulation field, while revealing many problems in the development of rural logistics [28]. For example, cost constraints in the process of rural circulation, inadequate function of rural logistics networks, and unbalanced logistics layout [29–31]. Li et al. (2020) [32] believed that the digital transformation of rural logistics was an inevitable trend in response to the changes of the times and a core component of quality development in the logistics industry. Kuihwa and park (2021) [33] analyzed the impact of the introduction of digital convergence technology on the existing logistics value chain, he considered digital convergence technology can secure the customer experience of logistics companies. Yin et al. (2020) [34] suggested that the new technological revolution could make up for the lack of rural logistics development by rationally introducing digital technology way and accelerating the construction of rural network facilities. Popkova and Sergi (2020) [35] based on conceptual scenario models and algorithms, concluded that digitization can maximize the effectiveness of transport and logistics.

Some scholars studied the relationship between the digital economy and the impact of rural logistics. Thanks to the innovative transformation of digital technology, which has led to a digital transformation of logistics, digital penetration has also been intensified in the rural logistics industry [36]. Big data, cloud computing, and other digital technologies provide more efficient and diversified information services for rural logistics [37]. Moreover, digital integration automates and digitizes logistics, which can continuously improve the input-output efficiency and development effectiveness of rural industries [38]. However, the digital capacity of rural logistics (the level of network facilities and services, the digitalisation process of industry, and the digital skills cultivation system) lags relatively behind, which limits the breadth and depth of digital technology penetration into rural agriculture [39]. After that, Zhang et al. (2019) [40] proposed that information processing capability was the foundation for building a digital shared information system, which was conducive to reshaping logistics business processes and innovating logistics operation models.

It remains a hot topic to research the path of coordinated development between digital economy and rural logistics. Aiming to improve the relatively lagging rural logistics industry,

the digital transformation path must be "adapted to local conditions" [41]. Digital technology can help eco-logistics expand the marketing channels for agricultural products and to achieve high-quality development of rural logistics [42]. Furthermore, Sun (2021) [43] considered that the deep integration of the new retail model and the modern logistics system would help to alleviate the contradiction of a "big market for small products". Leontev and Magera (2020) [44] suggested that rural logistics could consciously adjust the production structure and change the sales model, by improving the open, equal, and shared platform information and cooperation.

In summary, scholars have mainly analyzed the importance, the influential relationship, and the coordination path between the digital economy and rural logistics. It can be seen that the digital economy, as an important engine for China's high-quality economic development, is gradually integrating with the rural logistics industry, driving the rural logistics industry to make up for shortcomings, strengthen the foundation, digital transformation, and develop towards high quality. However, there is a lack of research on the harmonious relationship between them and the differences in regional development. One is that some studies still need to analyze in-depth, for example, the universality and specificity of the "data deficit" and its reasons for the high-quality development of the rural logistics industry. The second is that digital transformation is an important path for the rural logistics to break through the dilemma, but there is less research on the transformation path and relevance evaluation. The third is that a few studies point out that digital transformation is a core component of the high-quality development during the rural logistics, yet no study has systematically explored the logical relationship between the coordinated development of rural logistics and the digital economy, as well as regional differences.

Based on this, this study establishes an evaluation index system and constructs a model to evaluate the coupling and coordination relationship between the digital economy and the rural logistics, and further studies the differences in provincial development. To begin with, an analytical framework is established to clarify the logical relationship and operational structure of the coupled system, based on systems theory and coupling theory. Then, based on the analysis results of CD and CCD, the changing trends and regional variability are examined, and suggestions are provided for the coupled development between digital economy and rural logistics.

## Theoretical analysis of coupled system

According to the study of system theory [45] and coupling theory [43], this study takes the large system as the coupled system, which consists of the digital economy subsystem and the rural logistics subsystem. It is a complex that is coupled by the interaction of elements or behavioural forces within the system. The complex emphasises the holistic view of the system and forms a structurally coupled functional body, which has the characteristics of hierarchical complexity, structural relevance and synergistic evolution. Then, the coupled system includes both the material cycle and energy flow carried by the rural logistics system, and the integration of elements, information transfer and value flow between the two systems and within each sub-system driven by the digital economy system. As shown in Fig 1.

### Coupled system subjects and logical relationships

Digital economy and rural logistics are the two main subsystems that make up the coupling system. The digital economy subsystem is the sum of new economic forms, including a variety of new technologies, new products, new models, and new business forms based on digital technology. As well as economic growth is from the deep integration of digital elements with traditional industries [46]. The term "rural logistics subsystem" refers to the logistics services

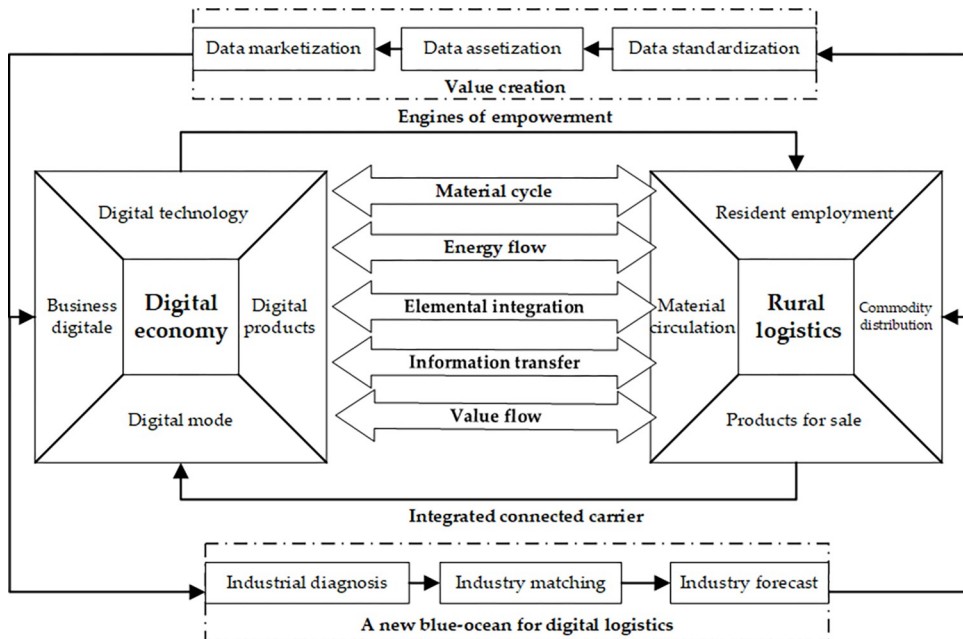

**Fig 1. Composition of the coupled system.**

offered to rural populations to support their production, living, and other economic activities [4]. It consists of a series of planning, implementation, and management of the entire process elements from the point of origin of commodities to the point of consumption of commodities in order to meet the supply of agricultural production materials, farmers' living and consumption needs, and the external circulation of agricultural products. And it is not only the basic hub linking the rural economy, linking production and marketing activities, but also a comprehensive coupling. Because of the mutual integration and influence between digital economy subsystem and rural logistics subsystem, the coupling system has the value creation function likes "marketization → capitalization → standardization" beyond the individual subsystem, and the potential of "industry diagnosis → industry matching → industry prediction", which finally forms the logical relationship network of the coupling system.

**Digital economy empowers rural logistics.** The digital economy, with data as a major factor of production, produces and develops economic forms such as digital industrialization and industrial digitization, and progressively becomes a source of power to promote high-quality economic development by gradually integrating with the actual economy [47]. The digital economy ecosystem, in particular, is a key engine for promoting high-quality agricultural and rural economic development [48], as well as the center of gravity for high-quality rural modernization, informatization, urbanization, and other "three rural" economies [49]. When it is difficult to obtain effective thrust from the old factors of production for the high-quality development in the rural logistics. The data factor become the new driving force, driving the formation of digital supply chains, digital logistics industry factors, and other digital transformation precedents [50]. Hence, which assistes rural logistics in forming a new model of "hard power and soft power" of production and digital integration. Then, it will cause the "four functional flows" (information flow, commercial flow, logistics, capital flow, and so on) and "five structural chains" (information chain, supply chain, value chain, technology chain, industrial chain, and so on) to undergo a digital "butterfly transformation", laying the groundwork for the supply of agricultural production materials, farmers' living and consumer demand, and the

external circulation of agricultural products [51]. In this regard, rural logistics has a larger logistics supply capacity and a more complete service system as a result of the digital economy's empowerment, expediting the rural logistics system's filling of gaps, laying a stable foundation, and promoting the development process.

**Rural logistics is the pivotal carrier of the digital economy.** The rural logistics becomes a complete system for material circulation, energy flow, information transfer, and value transfer in agriculture and rural areas that includes packaging, loading and unloading, distribution and processing, transportation, storage, and other methods [52]. Increased disposable income, improved network coverage, perfect "iron and public infrastructure" and "cloud network end" are clustered in the new countryside, which causes the market scale of rural logistics to expand. As an important means to stimulate rural vitality, the rural logistics promote the integrated development of rural industries, and is also the main causative factor in poverty-reduction [53]. To begin with, rural areas are critical for fostering and expanding the digital economy. The Communist Party of China's Central Committee has stated that it wants to "comprehensively promote 'Internet+' and create new advantages in the digital economy," and advocates for speeding up the exploration of China's regional industrial transformation and upgrading, as well as the innovative development of the digital economy [54]. Backward infrastructure, uneven network layout, limited marketability, and difficulties in comprehensive management are still problems, so the rural logistics makes a new field that has to be broken in the rural digital economy. Then, the development of the rural digital economy has prospects thanks to rural logistics. The logistics sector is integral to the functioning of the e-commerce platform created from the digital economy. At the same time, the digital economy also requires the integration from rural logistics, since it can support own high-quality development.

## Operational structure of the coupled system

The system may be separated into three interrelated and interconnected role structures with varying boundary ranges and functions based on the overall situation. The three layers are named the driving layer, the operation layer, and the quality control layer. The digital economy empowers the high-quality development of the rural economy, while rural logistics is the essential carrier of the rural digital transformation, based on the iterative function of the "three tiers" in this system. As shown in Fig 2.

**Driving layer: Systems' operation is supported by "two bases."** Supportive energy transformation is provided from the coupling and driving influence called the "two bases." On the one hand, the rural logistics industry, as a basic guarantee and key hub for agriculture and rural areas, primarily realizes the digital logistics transformation, through its role in living materials' employment and commodity distribution and agricultural products' large-scale production. On the other hand, the digital economy, which is based on digital technology, is gradually integrating with the rural logistics industry, is driven by a slew of digital innovation technologies, digital innovation products, digital business models, and other forms to build and develop a digital ecosystem for rural logistics. It allows rural logistics to become a high-quality development of the rural economy.

**Operation layer: Systems' goals are maintained and followed.** The operation layer mainly completes the "processing" operation in three aspects based on the energy input of the driving layer. Mismatched factors are eliminated such as backward infrastructure, uneven network layout, low marketability, and difficult comprehensive management. Furthermore, it promotes the digital and intelligent transformation that the rural logistics can form an efficient and refined system. This system is considered as "digitally driven and collaboratively shared". Subsequently, A new logistics system is being formed with digitalisation and platforming. It is

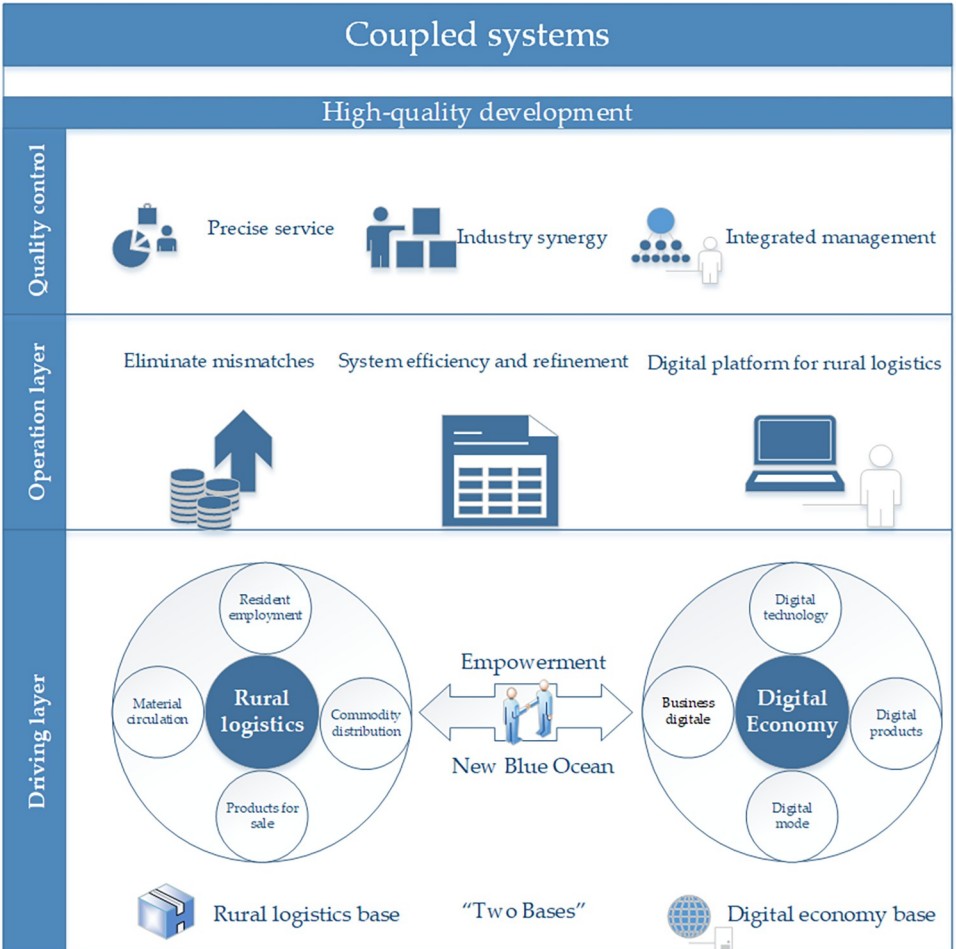

**Fig 2. Mechanism of action in the coupled system.**

helpful to compensate some of the drawbacks such as low level of digital applications, poor foundation of digital platforms, etc.

**Quality control layer: Elements are "picking" with the qualified goal.** Following the processing layer's activity, the coupling elements have developed a new paradigm of refinement, efficiency, digitalization, platform, and so on. The quality control layer's purpose is to cultivate a high-quality system that meets the requirements. System's characteristics keep comprehensive and precise services, industrial coordination and comprehensive management. There are in line with the orientation of the development of the coupling system composed of rural logistics and digital economy. These drive system to complete the efficiency change, power change, and quality change. Moreover, it has formed a high-quality rural digital logistics industry integrating digital production, on-demand processing, efficient distribution, and information technology services.

## Construction of the evaluation index system

In order to study the coupled and coordinated relationship between the digital economy subsystem and the rural logistics subsystem, the construction of an evaluation index system is important. With reference to Zhang et al. (2010) [55] and Zhang et al. (2020) [56], moreover,

following the principles of scientificity, rationality, comprehensiveness and accessibility in the selection of indicators, we have established an evaluation index system.

According to Wang (2014) [57], rural logistics is not only an important industry in rural areas. But also, an innovative ecosystem aims to integrate and optimize distribution resources and innovative business models, which seems an innovative network system for the circulation of rural materials. Therefore, the rural logistics is considered as a complex and multi-layered integrated system that follows the structural evolution of regional industrial self-organising systems. According to Chi (2015) [58], Zhang et al. (2018) [59], Ding and Wang (2019) [60], regional industrial ecosystems are mainly measured by using the perspectives of health, suitability and growth. At the meantime, the factors affecting the development of the rural logistics industry mainly include the scale of the industry, innovation resources, economic level, social living standard, policy environment, social environment and natural environment, in the process of the evolution of the rural logistics ecosystem. The results of these factors can be attributed to the high or low growth capacity of the rural logistics industry. Then, it requires corresponding capital investment such as technological advancement, model innovation, improvement of labour quality and construction of digital platforms in rural logistics. Therefore, this study chooses growth capacity and input support as the dimensions of rural logistics systems. Furthermore, according to Shang and Zhao (2021) [61], the industry growth capacity is classifyied into growth scale, growth capacity and growth environment specifically. Referring to Zhu et al. (2001) [62], indicator data is retrieved according to the criteria of the National Economic Classification of Industries. Data items are retrieved and collated corresponding from database resources such as China Statistical Yearbook, China Logistics Yearbook, China Rural Statistics Yearbook and China Logistics Development Report as evaluation indicators for the rural logistics subsystem. The whole comprehensive evaluation system includes 2 variables, 5 secondary dimensions and 11 evaluation indicators (e.g. A1-A2, B1-B9), as shown in Table 1.

Referring to the statistical measurement method of the Digital Economy Index by the China Electronics Information Industry Development Institute, digital economy activities mainly revolve around data collection, collation, analysis, security, distribution and

**Table 1. Evaluation index system for rural logistics subsystem.**

| Variables | Index number | Secondary Dimension | Index number | Calculated metrics | Unit | Index number |
|---|---|---|---|---|---|---|
| Input support | I | Fixed asset investment | $M_1$ | Investment in fixed assets in rural transportation, storage and postal industry | Hundred million RMB | $A_1$ |
| | | Human resource input | $M_2$ | Employment in transportation, storage and postal industry | Ten thousand people | $A_2$ |
| Growth capacity | G | Growth environment | $N_1$ | Rural residents | Ten thousand people | $B_1$ |
| | | | | Total agricultural output | Hundred million RMB | $B_2$ |
| | | | | Value added in agriculture as a percentage of regional GDP | KG | $B_3$ |
| | | Growth potential | $N_2$ | Disposable income of rural residents | RMB | $B_4$ |
| | | | | Consumption expenditure of rural residents | RMB | $B_5$ |
| | | | | The proportion of retail sales of consumer goods in town and village to the total retail sales of consumer goods | % | $B_6$ |
| | | Growth scale | $N_3$ | Cargo volume | Million tons | $B_7$ |
| | | | | Cargo Turnover | Hundred million tonnes/km | $B_8$ |
| | | | | Transportation routes | KM | $B_9$ |

consumption supported by digital technology [63]. This article selects a combination of five variables, based on their China Digital Economy Development Index (DEDI) Research Report (2017–2020). Namely, digital infrastructure, digital resources, digital technologies, and the impact of digital convergence and digital services that evolve on their own. Among them, the digital economy foundation is the "hard power" of the mid-level digital economy, which focuses on building the production and supply of digital goods or services, etc. Digital economy resources are regarded as "soft power", mainly because they comprise potential data resources and their application in the market economy. Digital economy technology consists of the input of cutting-edge and disruptive technologies in the digital economy, as well as the technology output resulting from the transfer and transformation of innovative technologies. The difference between digital economy convergence and digital services is more obvious, with the former mainly including the scale growth triggered by the convergence of primary and secondary industries, and the latter mainly referring to the portion of economic growth brought about by the convergence of digital technology and the tertiary industry. Then, combining the information from the Blue Book on Big Data: China's Big Data Development Report (2017–2020) and the White Paper on China's Digital Economy Index (2020), we get the evaluation system of the digital economy, including five dimensions and 28 evaluation indicators, as shown in Table 2. Finally, we extracted the statistics of DEDI from the China

**Table 2. Evaluation index system for digital economy subsystem.**

| Subsystem | Variables | Calculated metrics | Index number |
|---|---|---|---|
| Digital economy | Digital foundation | The scale of electronic information manufacturing industry | $C_1$ |
| | | Size of the information transmission industry | $C_2$ |
| | | Scale of Software and Information Technology Service Industry | $C_3$ |
| | | Average download rate of fixed broadband contracted broadband users | $C_4$ |
| | | Mobile Phone Penetration Rate | $C_5$ |
| | Digital resources | Number of big data enterprises listed | $D_1$ |
| | | Number of data trading centers | $D_2$ |
| | | Mobile Internet access traffic Number of mobile broadband subscribers | $D_3$ |
| | | Fixed Internet broadband access hours | $D_4$ |
| | | Number of fixed broadband subscribers | $D_5$ |
| | Digital technology | High-tech industry R&D personnel equivalent full-time equivalent | $E_1$ |
| | | Internal Expenditure on R&D in High-tech Industry | $E_2$ |
| | | Patents in the high-tech industry | $E_3$ |
| | | High-tech industry technology acquisition and industrial transformation expenditures | $E_4$ |
| | Digital convergence | Number of agricultural Internet platforms | $F_1$ |
| | | Percentage of enterprises with e-commerce trading activities | $F_2$ |
| | | Number of National Demonstration Enterprises for Integration of Two | $F_3$ |
| | | Penetration rate of digital R&D and design tools | $F_4$ |
| | | Numerical control rate of key processes Intelligent manufacturing readiness rate | $F_5$ |
| | Digital services | Instant communication-WeChat user distribution | $G_1$ |
| | | Travel—trip user distribution | $G_2$ |
| | | Life service-new media user distribution | $G_3$ |
| | | Online shopping-Net retail sales | $G_4$ |
| | | Internet finance-Alipay user distribution | $G_5$ |
| | | Entertainment-akiyee user distribution | $G_6$ |
| | | Education-Internet access rate in primary and secondary schools | $G_7$ |
| | | Internet healthcare—Ping an good doctor user distribution | $G_8$ |
| | | Travel-Drip travel user distribution | $G_9$ |

Digital Economy Development Index (DEDI) Research Report (2017–2020), and used DEDI as the measurement data of the digital economy subsystem.

## Model construction

A coupling coordination model is constructed to verify the coupling relationship between the digital economy subsystem and the rural logistics subsystem. In this subsection, subsystems' evaluation functions can be derived. Then, the CD and CCD functions are used to build the coupling coordination degree model. Therefrom, it provides data and model support for the research and analysis of coupling relationships. The specific research process is shown in Fig 3.

### Comprehensive evaluation functions

**Evaluation function of rural logistics subsystem.** In a system of multiple indicators, determining indicator weights is essential [64]. The methods to determine weights mainly

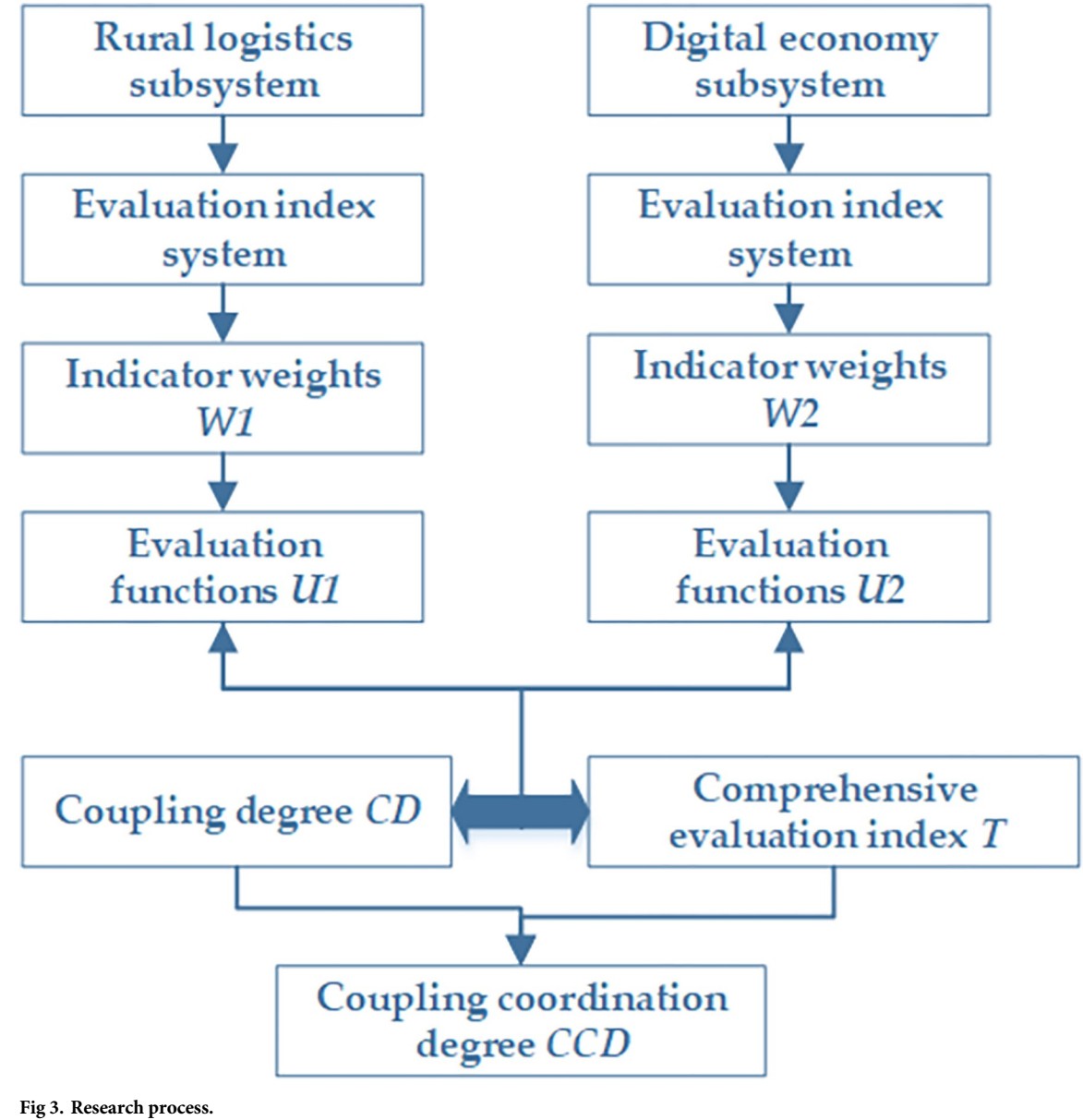

**Fig 3. Research process.**

include subjective, objective, and combination. As the indicator data possesses characteristics such as disorder and dispersion. Compared with subjective weighting methods such as Delphi, hierarchical analysis, and principal component analysis, the entropy weighting method can make the results objective, highly credible, and easy to operate. Therefore, it is chosen to measure for rural logistics subsystem, and the specific steps are as follows.

1. Normalisation of the evaluation matrix

The $n$ evaluation indicators from the $m$ samples are arranged in order into the original data matrix $X = (x_{\phi_{ij}})_{m \times n}$, $x_{\phi_{ij}}$ which is the raw data value of the $\phi$ indicator for province $i$ in year $j$.

In order to eliminate the influence of the scale of each indicator, its own variability and the size of the values under the subsystem, the data keeps standardized using the extreme difference standardization [65].

For positive indicators, order.

$$X_{\phi_{ij}}{'} = \frac{x_{\phi_{1j}} - Min(x_{\phi_{1j}}, x_{\phi_{2j}} \ldots x_{\phi_{mj}})}{Max(x_{\phi_{1j}}, x_{\phi_{2j}} \ldots x_{\phi_{mj}}) - Min(x_{\phi_{1j}}, x_{\phi_{2j}} \ldots x_{\phi_{mj}})} \times 0.9 + 0.1 \tag{1}$$

For negative indicators, order.

$$X_{\phi_{ij}}{'} = \frac{Max(x_{\phi_{1j}}, x_{\phi_{2j}} \ldots x_{\phi_{mj}})}{Max(x_{\phi_{1j}}, x_{\phi_{2j}} \ldots x_{\phi_{mj}}) - Min(x_{\phi_{1j}}, x_{\phi_{2j}} \ldots x_{\phi_{mj}})} \times 0.9 + 0.1 \tag{2}$$

Where $x_{\phi_{ij}}{'}$ represents standardization of the $\phi$ indicator for $i$ in year $j$.

2. Calculation of entropy weights

According to the standardized data, the weighting matrix is calculated for each indicator. As shown in Eq (3).

$$(P_{ij})_{m \times n} = \frac{X_{\phi_{ij}}{'}}{\sum_{i=1}^{m} X_{\phi_{ij}}{'}} \ (j = 1, 2, 3 \ldots n; 0 \le P_{ij} \le 1) \tag{3}$$

The entropy value of each indicator is shown in Eq (4). where is the entropy coefficient (entropy is a measure in thermodynamic systems. The lower the entropy, the more orderly the self-organization).

$$H_j = -\frac{1}{\ln n} \sum_{i=1}^{n} P_{ij} \ln P_{ij} \tag{4}$$

The entropy weight is calculated as shown in (5), where $H_j$ is between 0 and 1, and $w_j$ represents the entropy weight. Therefore, the smaller the degree of variation of the resulting evaluation index, the greater the entropy value.

$$W_j = \frac{1 - H_j}{n - \sum_{j=1}^{n} H_j} \tag{5}$$

## Coupling coordination model and evaluation criteria

The coupling derives from the coupling principle in the physical discipline, which refers to the synergistic effect of two or more self-organizing systems or forms of motion interacting with each other [64]. In this paper, we use the CD to describe how the two subsystems interact with each other. Synthesizing a series of formulas for the coupling degree model, given $n \ge 2$ systems and denoting the evaluation value of a subsystem by $U_i \ge 0$. According to Cong (2019) [65], the

expression of the coupling degree model generally takes two forms, as follows (6) and (7).

$$C_1(U_1, U_2, \ldots U_n) = n \times \left[\frac{U_1, U_2, \ldots U_n}{(U_1 + U_2 + \ldots + U_n)^n}\right]^{\frac{1}{n}} \tag{6}$$

$$C_1(U_1, U_2, \ldots U_n) = 2 \times \left[\frac{U_1, U_2, \ldots U_n}{\prod (U_i + U_j)^{\frac{2}{n-1}}}\right]^{\frac{1}{n}} \tag{7}$$

When $n = 2$, $0 \leq C_1 = C_2 \leq 1$, according to the basic properties of the coupling degree model. Thus, we set the overall combined contribution of the indicators in the rural logistics subsystem to $U_1$ and the overall combined contribution of the indicators in the digital economy subsystem to $U_2$, which leads to the coupling degree (CD) expression for the coupled system as in Eq (8).

$$C = 2 \times \left[\frac{U_1 U_2}{(U_1 + U_2)^2}\right]^{\frac{1}{2}} \tag{8}$$

However, CD alone is not sufficient, which reflects the strength of the interaction between subsystems. Some scholars believe that, if the coupled system has a high CD but a low overall level of development, this will lead to a bias in the coordination of the system [66, 67]. Yet, CCD reflects the level of coordinated development and the degree of harmony and coherence. Therefore, CCD can be a measure required during the developmental evolutionary process. The measurement Eq is as follows (9).

$$\begin{cases} D = \sqrt{C \times T} \\ T = aU_1 + bU_2 \end{cases} \tag{9}$$

According to Eq (9), $D$ is the CCD, and $T$ is the comprehensive evaluation index of the coordination effect. Then, $a$ and $b$ denote the weight coefficients in subsystems. With reference to Wu(2016) [68], We set $a = 0.4$ and $b = 0.6$, are taken to make $T \in [0.0, 1.0]$ and $D \in [0.0, 1.0]$.

Referring to Cong (2019) [65], when only two subsystems are included in a large system, Eqs (10) and (11) are equal. That is, when $U_1 = U_2$, $CD = 1$. Based on this characteristic, we divide CD into seven intervals, as shown in Table 3.

Similarly, by the median partitioning method, the CCD can be partitioned into 10 evaluation intervals [69], as shown in Table 4.

**Table 3. Evaluation criteria for CD.**

| CD range | Coupling type |
|---|---|
| $CD = 0$ | System disorder (Each subsystem is in an independent state, and the system develops into disorder) |
| $0 < CD \leq 0.5$ | Low-level coupling |
| $0.5 < CD \leq 0.7$ | Antagonistic stage |
| $0.7 < CD \leq 0.9$ | Running in stage |
| $0.9 < CD \leq 0.95$ | Good level coupling |
| $0.95 < CD < 1$ | High-level coupling |
| $CD = 1$ | Full Coupling (In the state of benign resonance coupling, the system develops orderly) |

**Table 4. Evaluation criteria for CCD.**

| CCD range | CCD type | Level | CCD range | CCD type | Level |
|---|---|---|---|---|---|
| (0.0,0.1] | Extreme maladjustment | 1 | (0.5,0.6] | Grudging coordination | 6 |
| (0.1,0.2] | Severe maladjustment | 2 | (0.6,0.7] | Primary coordination | 7 |
| (0.2,0.3] | Moderate maladjustment | 3 | (0.7,0.8] | Intermediate coordination | 8 |
| (0.3,0.4] | Mild maladjustment | 4 | (0.8,0.9] | Good coordination | 9 |
| (0.4,0.5] | On the verge of maladjustment | 5 | (0.9, 1.0] | High-quality coordination | 10 |

## Relationship evaluation and analysis

### Data processing

21 provinces are selected for the study, such as Beijing and Tianjin in China. The data are for the period 2016 to 2019. Due to the lack of measured data on the rural logistics industry, such as Mongolia, Tibet, and Qinghai, etc. Then, reference is made to Yi [70], Executive Director of the China Institute of Modern Economics, who considered 2016 to be the first year of China's digital economy. In addition, according to the statistics, the data has only been more complete since 2016, for the indicator data of the digital economy. Therefore, this study has chosen the period of 2016–2019 as the time frame for the research sample.

**Calculation of indicator weights.** The entropy weighting method is applied to determine the indicator weights for the rural logistics subsystem after standardizing the assessment matrix. As is shown in Table 5.

**Calculation of the evaluation index.** According to Eq (3–5), the evaluation index is got in the rural logistics subsystem. Referring to the DEDI released by the China Electronics Information Industry Development Institute, the evaluation index in the digital economy subsystem is also obtained. Accordingly, evaluation values are integrated in the coupled system, as shown in Table 6.

**Harris-Tzavalis unit-root test.** The stability of the data needs to be checked to prevent "pseudo-regressions", and to guarantee that the findings are statistically legitimate. Therefore,

**Table 5. Indicator weights for the rural logistics subsystem.**

| Index number | 2016 | 2017 | 2018 | 2019 |
|---|---|---|---|---|
| A1 | 0.6505 | 0.6940 | 0.6228 | 0.6636 |
| A2 | 0.3495 | 0.3060 | 0.3772 | 0.3364 |
| B1 | 0.3663 | 0.2809 | 0.3648 | 0.3638 |
| B2 | 0.3776 | 0.2939 | 0.3619 | 0.3560 |
| B3 | 0.2561 | 0.4252 | 0.2733 | 0.2802 |
| B4 | 0.4352 | 0.4539 | 0.4413 | 0.4766 |
| B5 | 0.3881 | 0.3826 | 0.4029 | 0.3829 |
| B6 | 0.1767 | 0.1635 | 0.1558 | 0.1405 |
| B7 | 0.4038 | 0.2724 | 0.2824 | 0.2915 |
| B8 | 0.3627 | 0.4337 | 0.4184 | 0.4167 |
| B9 | 0.2335 | 0.2939 | 0.2992 | 0.2918 |
| M1 | 0.6505 | 0.6940 | 0.6228 | 0.6636 |
| M2 | 0.3495 | 0.3060 | 0.3772 | 0.3364 |
| N1 | 0.4089 | 0.4230 | 0.4029 | 0.3960 |
| N2 | 0.2848 | 0.2982 | 0.3267 | 0.3341 |
| N3 | 0.3063 | 0.2788 | 0.2704 | 0.2699 |

**Table 6. Evaluation values for coupled systems.**

| province | 2016 | | 2017 | | 2018 | | 2019 | |
|---|---|---|---|---|---|---|---|---|
| | $U_1$ | $U_2$ | $U_1$ | $U_2$ | $U_1$ | $U_2$ | $U_1$ | $U_2$ |
| Beijing | 0.371 | 0.713 | 0.359 | 0.660 | 0.403 | 0.772 | 0.397 | 1.000 |
| Tianjing | 0.284 | 0.377 | 0.241 | 0.346 | 0.233 | 0.209 | 0.227 | 0.285 |
| Hebei | 0.501 | 0.375 | 0.466 | 0.380 | 0.657 | 0.259 | 0.556 | 0.392 |
| Shanxi | 0.558 | 0.154 | 0.515 | 0.228 | 0.546 | 0.200 | 0.390 | 0.195 |
| Liaoning | 0.420 | 0.337 | 0.347 | 0.350 | 0.382 | 0.325 | 0.381 | 0.252 |
| Jilin | 0.286 | 0.208 | 0.248 | 0.224 | 0.240 | 0.180 | 0.205 | 0.107 |
| Heilongjiang | 0.370 | 0.148 | 0.320 | 0.164 | 0.324 | 0.191 | 0.375 | 0.181 |
| Shanghai | 0.466 | 0.630 | 0.462 | 0.653 | 0.643 | 0.704 | 0.418 | 0.774 |
| Jiangsu | 0.525 | 1.000 | 0.456 | 1.000 | 0.514 | 0.765 | 0.480 | 0.934 |
| Zhejiang | 0.413 | 0.883 | 0.340 | 0.869 | 0.542 | 0.670 | 0.432 | 0.917 |
| Anhui | 0.484 | 0.421 | 0.318 | 0.442 | 0.399 | 0.355 | 0.444 | 0.390 |
| Fujian | 0.355 | 0.556 | 0.284 | 0.552 | 0.598 | 0.524 | 0.320 | 0.611 |
| Jiangxi | 0.343 | 0.235 | 0.332 | 0.231 | 0.332 | 0.200 | 0.294 | 0.371 |
| Henan | 0.547 | 0.405 | 0.479 | 0.431 | 0.565 | 0.394 | 0.826 | 0.525 |
| Hubei | 0.744 | 0.472 | 0.747 | 0.444 | 0.726 | 0.385 | 0.401 | 0.466 |
| Guangxi | 0.539 | 0.138 | 0.455 | 0.208 | 0.534 | 1.000 | 0.500 | 0.316 |
| Chongqing | 0.284 | 0.303 | 0.240 | 0.291 | 0.374 | 0.289 | 0.314 | 0.378 |
| Sichuan | 0.739 | 0.484 | 0.581 | 0.445 | 0.654 | 0.489 | 0.403 | 0.539 |
| Yunnan | 0.000 | 0.350 | 0.002 | 0.462 | 0.063 | 0.393 | 0.111 | 0.232 |
| Shanxi | 0.228 | 0.485 | 0.171 | 0.601 | 0.152 | 0.686 | 0.243 | 0.495 |
| Ningxia | 0.018 | 0.285 | 0.000 | 0.316 | 0.000 | 0.401 | 0.000 | 0.130 |

a unit root test on the panel data is implemented by using the Stata13 program. And the Harris-Tzavalis (1999) [71] test is chosen, considering that the sample data had a feature (N>T). The initial statement of the hypothesis was "Ho: Panels contain unit roots." When the initial theory is disproven, it is stated as "Ha: Panels are stationary." Furthermore, Y and X are set as the evaluation indices for the rural logistics subsystem and the digital economy subsystem respectively. And X and Y underwent the Harris-Tzavalis unit root test. Table 7 displays the results.

According to the results in Table 7, the test results of the evaluation indices (X and Y) of the rural logistics subsystem and the digital economy subsystem both reject the original hypothesis. Therefore, both X and Y are stable series, and the coupling relationship can be measured.

## Evaluation analysis of the CD and coupling type

Based on the evaluation criteria of CD, this section has evaluated and analyzed three aspects respectively. To begin with, from 2016 to 2019, the CD shows from 0.784 to 1, indicating that they have a coupled relationship of mutual feedback and interaction. Importantly, in 2017,

**Table 7. Harris-Tzavalis unit-root test for Y and X.**

| Subsystems | Variables | Statistic | Z | P-value | Test results | Stabilization |
|---|---|---|---|---|---|---|
| Rural logistics | X | -0.3021 | -5.5640 | 0.0000*** | Reject Ho | Yes |
| Digital economy | Y | -0.0686 | -3.7133 | 0.0001*** | Reject Ho | Yes |

Note: *** denotes significance at 1% confidence level

**Table 8. CD and coupling taye.**

| province | 2016 | | 2017 | | 2018 | | 2019 | |
|---|---|---|---|---|---|---|---|---|
| | CD | Coupling type | CD | Coupling type | CD | Coupling type | CD | Coupling type |
| Beijing | 0.949 | Good level coupling | 0.955 | High-level coupling | 0.949 | Good level coupling | 0.902 | Good level coupling |
| Tianjin | 0.990 | High-level coupling | 0.984 | High-level coupling | 0.998 | High-level coupling | 0.994 | High-level coupling |
| Hebei | 0.990 | High level coupling | 0.995 | High-level coupling | 0.900 | Running in stage | 0.985 | High-level coupling |
| Shanxi | 0.824 | Running in stage | 0.922 | Good level coupling | 0.886 | Running in stage | 0.943 | Good level coupling |
| Liaoning | 0.994 | High-level coupling | 1.000 | Full Coupling | 0.997 | High-level coupling | 0.979 | High-level coupling |
| Jilin | 0.988 | High-level coupling | 0.999 | High-level coupling | 0.990 | High-level coupling | 0.950 | Good level coupling |
| Heilongjiang | 0.904 | Good level coupling | 0.946 | High-level coupling | 0.966 | High-level coupling | 0.937 | Good level coupling |
| Shanghai | 0.989 | High-level coupling | 0.985 | High-level coupling | 0.999 | High-level coupling | 0.954 | High-level coupling |
| Jiangsu | 0.950 | Good level coupling | 0.927 | Good level coupling | 0.981 | High-level coupling | 0.947 | Good level coupling |
| Zhejiang | 0.932 | Good level coupling | 0.899 | Running in stage | 0.994 | High-level coupling | 0.933 | Good level coupling |
| Anhui | 0.998 | High-level coupling | 0.987 | High-level coupling | 0.998 | High-level coupling | 0.998 | High-level coupling |
| Fujian | 0.975 | High-level coupling | 0.947 | Good level coupling | 0.998 | High-level coupling | 0.950 | Good level coupling |
| Jiangxi | 0.982 | High-level coupling | 0.984 | High-level coupling | 0.969 | High-level coupling | 0.993 | High-level coupling |
| Henan | 0.989 | High-level coupling | 0.999 | High-level coupling | 0.984 | High-level coupling | 0.975 | High-level coupling |
| Hubei | 0.975 | High-level coupling | 0.967 | High-level coupling | 0.952 | High-level coupling | 0.997 | High-level coupling |
| Guangxi | 0.806 | Running in stage | 0.928 | Good level coupling | 0.953 | High-level coupling | 0.974 | High-level coupling |
| Chongqing | 0.999 | High-level coupling | 0.995 | High-level coupling | 0.992 | High-level coupling | 0.996 | High-level coupling |
| Sichuan | 0.978 | High-level coupling | 0.991 | High-level coupling | 0.989 | High-level coupling | 0.989 | High-level coupling |
| Yunnan | 0.784 | Running in stage | 0.785 | Running in stage | 0.872 | Running in stage | 0.957 | High-level coupling |
| Shanxi | 0.996 | High-level coupling | 0.961 | High-level coupling | 0.935 | Good level coupling | 0.989 | High-level coupling |
| Ningxia | 0.938 | Good level coupling | 0.914 | Good level coupling | 0.833 | Running in stage | 0.972 | High-level coupling |
| **Mean** | **0.949** | Good level coupling | **0.956** | High-level coupling | **0.959** | High-level coupling | **0.967** | High-level coupling |

Liaoning had the highest CD, indicating that in Liaoning province, the rural logistics subsystem is the most coupled to the digital economy subsystem. Immediately after, in the vertical analysis, the mean of CD tends to increase year by year. From 2016 to 2017, the coupling system crossed from the better coupling stage to the high coupling level, even in 2019, the CD of all 21 provinces were greater than 0.9, and the provinces were located at the better coupling or high coupling level. However, the relationship between two subsystems in several provinces was at an unstable stage from 2016 to 2018, for example, the CD in Yunnan province were all at the grinding stage level, and five provinces (Hebei, Shanxi, Zhejiang, Guangxi, and Ningxia) intermittently presented the grinding stage. Finally, in the cross-sectional analysis, there are differences in the stability of the coupling system across provinces from 2016 to 2019. Among them, eight provinces, including Tianjin, Shanghai, Anhui, et al., all have coupled systems at a high coupling level. More notably, Liaoning Province has a CD value of 1 in 2017, reaching a fully coupled level. As shown in Table 8.

## Evaluation analysis of CCD and coordination levels

Overall, CCD ranges from 0.348 to 0.877, indicating that the coupled system was located between mildly dysfunctional and well-coordinated from 2016 to 2019. Among them, Jiangsu had the highest CCD in 2016 (0.877) and Ningxia had the lowest CCD in 2019 (0.348). To explore the coupling and coordination patterns of the two subsystems, 21 provinces are divided into four echelons, as shown in Table 9.

Jiangsu, Zhejiang, Beijing, and Shanghai form the first echelon. The CCD runs in the range of 0.718 to 0.877, with the coupled systems evolving between primary to good coordination.

**Table 9. CCD and coordination level.**

| Province | 2016 | | 2017 | | 2018 | | 2019 | | Coordinated echelons |
|---|---|---|---|---|---|---|---|---|---|
| | CCD | Level | CCD | Level | CCD | Level | CCD | Level | |
| Jiangsu | 0.877 | 9 | 0.852 | 9 | 0.807 | 9 | 0.844 | 9 | **First echelon** |
| Zhejiang | 0.805 | 9 | 0.769 | 8 | 0.785 | 8 | 0.821 | 9 | |
| Beijing | 0.739 | 8 | 0.718 | 8 | 0.770 | 8 | 0.827 | 9 | |
| Shanghai | 0.747 | 8 | 0.754 | 7 | 0.824 | 9 | 0.777 | 8 | |
| $V_{CCD1}$ | **0.792** | **8** | **0.773** | **8** | **0.796** | **8** | **0.817** | **9** | |
| Sichuan | 0.757 | 8 | 0.704 | 7 | 0.741 | 8 | 0.693 | 7 | **Second echelon** |
| Hubei | 0.752 | 8 | 0.739 | 8 | 0.705 | 7 | 0.662 | 7 | |
| Fujian | 0.681 | 7 | 0.649 | 7 | 0.743 | 8 | 0.685 | 7 | |
| Henan | 0.676 | 7 | 0.671 | 7 | 0.675 | 7 | 0.793 | 8 | |
| Anhui | 0.667 | 7 | 0.622 | 7 | 0.610 | 7 | 0.641 | 7 | |
| Hebei | 0.649 | 7 | 0.642 | 7 | 0.613 | 7 | 0.671 | 7 | |
| $V_{CCD2}$ | **0.697** | **7** | **0.671** | **7** | **0.681** | **7** | **0.691** | **7** | |
| Liaoning | 0.607 | 7 | 0.590 | 6 | 0.589 | 6 | 0.545 | 6 | **Third echelon** |
| Chongqing | 0.543 | 6 | 0.519 | 6 | 0.566 | 6 | 0.592 | 6 | |
| Shanxi | 0.510 | 6 | 0.562 | 6 | 0.547 | 6 | 0.507 | 6 | |
| Shanxi | 0.573 | 6 | 0.565 | 6 | 0.565 | 6 | 0.598 | 6 | |
| Jiangxi | 0.522 | 6 | 0.517 | 6 | 0.495 | 5 | 0.581 | 6 | |
| Tianjing | 0.580 | 6 | 0.547 | 6 | 0.467 | 5 | 0.510 | 6 | |
| Guangxi | 0.490 | 5 | 0.534 | 6 | 0.880 | 9 | 0.616 | 7 | |
| $V_{CCD3}$ | **0.547** | **6** | **0.548** | **6** | **0.587** | **6** | **0.564** | **6** | |
| Heilongjiang | 0.463 | 5 | 0.463 | 5 | 0.486 | 5 | 0.492 | 5 | **Fourth echelon** |
| Yunnan | 0.426 | 5 | 0.429 | 5 | 0.492 | 5 | 0.504 | 5 | |
| Jilin | 0.486 | 5 | 0.483 | 5 | 0.449 | 5 | 0.373 | 4 | |
| Ningxia | 0.394 | 4 | 0.376 | 4 | 0.407 | 4 | 0.348 | 4 | |
| $V_{CCD4}$ | **0.442** | **5** | **0.438** | **5** | **0.458** | **5** | **0.429** | **5** | |

Furthermore, in terms of mean values, the CCD keeps greater than 0.7, forming a leap from intermediate coordination to good coordination. Six provinces, including Sichuan, Hubei, and Fujian, etc. form the second echelon and made a stable development pattern. And CCD lies from 0.610 to 0.793, its type evolves from primary coordination to intermediate coordination. Then, seven provinces, including Liaoning, Chongqing, and Shanxi, etc. show a third echelon pattern of left-right resistance. The mean keeps greater than 0.5 ($0.547 \leq V_{CCD3} \leq 0.587$), which makes grudging coordination. Finally, Heilongjiang, Yunnan, Jilin, and Ningxia show a fourth echelon pattern of coupling dissonance. Which is in a low coordination zone, evolving between mild maladjustment and on the verge of maladjustment.

**Convergence analysis of coupled and coordinated development.** Due to the differences in CCD across the echelons, this study performs convergence analysis tests. The aim is to further verify the coordinated evolutionary characteristics of the two subsystems. Referring to Xing et al. (2019) [72], $\sigma$ convergence analysis is applied in each echelon. The test formula is as follows.

$$\sigma_t = \left\{ N^{-1} \sum\nolimits_{c=1}^{N} \left[ X_c(t) - \left[ N^{-1} \sum\nolimits_{k=1}^{n} X_k(t) \right] \right]^2 \right\}^{\frac{1}{2}} \tag{10}$$

Where, $X_c(t)$ denotes the CCD of the $c$ province in year $t$, and $N$ represents the number of provinces. If $\sigma_{t+1} \leq \sigma_t$, then it means that the gap in each echelon is gradually decreasing. The measurement results are shown in Fig 4.

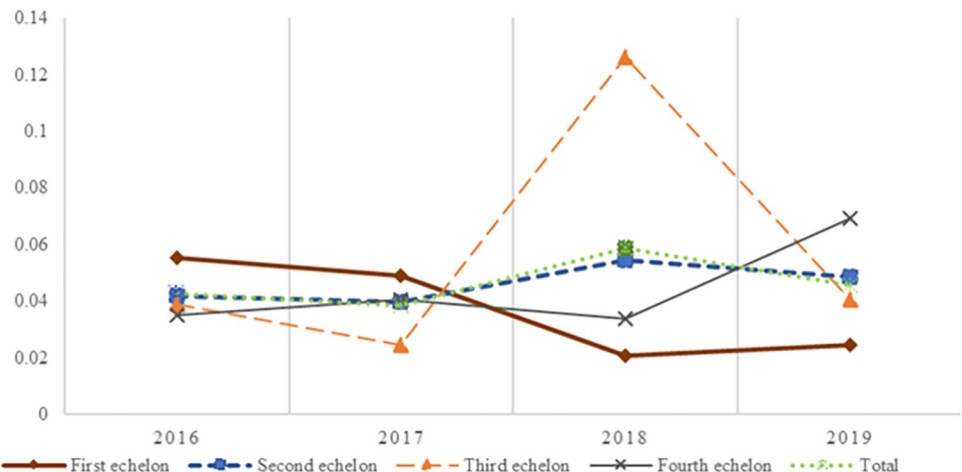

**Fig 4. The CCD development trends of the echelons.**

As can be seen from Fig 4, the coefficient of variation is small (all less than 0.2) in 21 provinces. And it forms an inverted U-shaped convergence trend through alternating fluctuations of increase and decreases, which indicates that the gap is gradually decreasing. In addition, the trends in CCD evolution vary across the echelons. The first echelon shows small fluctuations, although the overall coefficient of variation shows a gradually decreasing trend. The second echelon and the 21 provinces have almost similar trends, both showing an inverted U-shaped. The third echelon shows more obvious alternating characteristics of increasing and decreasing, and produce a more obvious bump in 2018. The fourth echelon shows a non-converging trend from decreasing to increasing, indicating that it is different for the CCD of the coupled system.

## Conclusions

In the process of building a new socialist countryside in China, rural logistics is seen as a bridge link, an effective support and important guarantee. As the digital factor rises to become a new factor of production, it is driving both a change in the development of the rural economy and an opportunity for the high-quality development of rural logistics. The coupling of the digital economy and rural logistics has become a breakthrough in development. In this context, this article elucidates the subject's logical relationships as well as the operational structure in coupled systems, based on theoretical perspectives such as systems theory and coupling theory. Followed by, the coupling coordination model is constructed to verify coupled relationships as well as the differences in coupled coordination, etc. Then, after convergence analysis, findings show that the coupling keeps dynamic and there is variability in the four echelons. Therefore, the findings answer three questions well. First, the operating structure of the coupling system is well-analyzed and elaborated. Second, taking the development of 21 China's provinces as the research object, the results show that there is a coupling relationship between the digital economy subsystem and the rural logistics subsystem. Third, the evolution of coupled systems is dynamic. More importantly, in different periods, 21 provinces have different stages of coupling and coordination, reflecting the differences in development in time and space.

All in all, this study embodies innovation in both theory and practice. On the theoretical side, the research scope of coupling theory has been expanded. Combining systems theory with coupling theory, we have studied the logical structure, coupled relationships, and development variability. On the practical side, we enrich the application practice between rural

logistics and the digital economy. Firstly, we analyze their coupling degree, coupling evolution stages, and coupling evolution trends using the coupling coordination model. Secondly, as examples for the digital transformation of rural logistics in each province, we also suggest policy ideas for the execution of high-quality development for various echelons.

This study gets innovative in both theory and application. However, there are still certain restrictions because of factors like time and data. The system for evaluating indicators still has flaws. The qualitative dimension has not been fully accounted for by the system of indicators used to assess subsystems. Even though the selection procedure is strictly followed the paradigm. The problem of indicator bias is not taken into account and has to be calibrated and improved in various settings or provinces. Next, this study analyzes the CD and CCD in the system using panel data as a sample without considering the dynamic, ongoing effects of the coupled system. Further research on more in-depth research topics may include continual data gathering and matching, in our opinion. At last, provinces and years considered in this work have limits given the availability of data. The follow-up study has to be improved further to try to correct the sample size bias in the results.

## Discussion

This study offers a new perspective on the literature in several ways. The first is that we expand on the theoretical aspect. We combine systems theory [44] with coupling theory [45] to elucidate the subjective logical relationships as well as the operational structure of a coupled system, which is composed by a digital economy subsystem and a rural logistics subsystem. Furthermore, current research has mentioned that it is an inevitable trend regarding the integration of the digital economy and rural logistics. However, existing research has mainly focused on analyzing the importance, and impact relationships, and coordination paths between the digital economy and rural logistics. There is still room for a breakthrough in terms of clarifying the coupling and coordination between rural logistics and the digital economy, as well as regional differences. To fill the gap in the study of the relationship between rural logistics and the digital economy, referring to Li, et al. (2021) [21], a coupled coordination model is applied to evaluate the CD and CCD. Based on the evaluation criteria of CD and CCD, findings indicate that there is a coupled relationship between rural logistics and the digital economy with mutual feedback and mutual influence. It shows that the coupling system driven by two subsystems develops from low to high level, and their relationship has improved positively.

In addition, because two subsystems are in evolutionary development from mild maladjustment to high-quality coordination. It indicates the digital economy and rural logistics maintain a coordinating relationship. At the same time, according to the evolutionary pattern in different years and provinces, four echelons have been identified according to the evolution of coupled coordination. Among them, four provinces, namely Jiangsu, Zhejiang, Beijing and Shanghai, form the first echelon pattern. Six provinces, including Sichuan, Hubei, Fujian, etc., form the steadily driving second echelon pattern. Seven provinces, including Liaoning, Chongqing, etc., show the third echelon pattern of left-right resistance. Four provinces, including Heilongjiang, Yunnan, Jilin and Ningxia, show the fourth echelon pattern of coupling disorder.

Last but not least, from the perspective of development variability, two conclusions can be drawn. Firstly, in terms of long-term development trends, the coefficient of variation of the CCDs in the 21 provinces is small ($\sigma$ less than 0.2), and through the fluctuating changes of alternating increases and decreases, an inverted U-shaped convergence trend is formed. This indicates that the gap keeps a gradual decrease. Secondly, taking each echelon as the object of examination, the trend seems to differ. The σ values of the first echelon coupled systems show

a slow convergence. The second echelon has an almost similar trend with 21 provinces, which presents an inverted U-shaped convergence trend. The third echelon displays more obvious alternating characteristics of increase and decrease. The fourth echelon also shows a non-converging trend of decrease followed by an increase with 2018 as the cut-off point.

Based on the above analysis, we make some points. The evolution of coordination is inextricably linked to the basis of economic development. Based on the development characteristics of the Chinese provinces, we can well contrast practical and theoretical research. The details are as follows.

1. Four provinces make up the first echelon. Jiangsu in the first place, has the best-coordinated development, which is supported by advantageous factors such as a well-developed logistics infrastructure network driven by digital research and development capabilities, and broad market space. Due to the high concentration of data centers and the larger Taobao operator base, which serves as the foundation for data technology and logistics applications for the digital transformation of rural logistics, Zhejiang and Shanghai are close behind. Beijing, on the other hand, has concentrated on building out the infrastructure for the digital economy since 2016, gradually developing a significant digital economy sector in the Beijing-Tianjin-Hebei region.

2. Six provinces form a steadily driving second echelon. First, while using resources from building the Yangtze River Economic Belt, Sichuan and Hubei are constantly improving the fundamental environment in rural logistics. Then, since it builds on the foundation of regional markets and information technology, Fujian has a distinct edge in the integration of the digital economy and rural logistics. Additionally, based on the national strategy for ecological conservation and high-quality development, Henan pushes for further optimization. Finally, Although Anhui and Hebei are only at the initial coordination stage, their potential for coupling in the future is enabled by the peculiarities of their respective provincial industries.

3. There are seven provinces forming the third echelon, showing a left-right counterweight. To start with, Liaoning, Chongqing, Shanxi, and Shaanxi maintain a barely harmonized level, as they strive for a breakthrough in the track of quality development of rural logistics. Next, Jiangxi province, which relies on the dividends of the construction of the Yangtze River Economic Belt, and Tianjin is known as the "birthplace of modern industry". Although they have already laid out their digital strategies in rural logistics, the coupling system is at risk of becoming out of tune in 2018 due to geographical and other factors such as the economic development base. Last but not least, Guangxi fluctuates the most during the period under examination (from 0.49 to 0.88). As a port in the western region, which still needs to build up its strength in terms of stability, although it is also easier to gather development factors such as manpower, materials, and information than others.

4. Four provinces make up the fourth echelon, which seems characteristics of coupling disorder. The four provinces are taking a while to establish the integration of the digital economy and rural logistics because of economic, geographic, and historical constraints. Additionally, due to their significant gaps in e-commerce applications, digital financial services, and the development of logistics systems, rural areas experience issues with spatial and temporal limits in their resource flows and information exchanges.

## Suggestions

Some adaptive suggestions are made for the coupling development between digital economy and rural logistics in each province.

### Consolidating the path of coordination and integration

From a comprehensive perspective, provinces need to solidify the development path of coordination and integration, to generate new paradigms and dynamics of digital integration. The

two subsystems keep coupled and coordinated relationship with mutual feedback and influence. However, the coupled system mainly consists of the digital economy and rural logistics, is undergoing a complex evolution from low to a high level. Meantime, there are still difficulties in the coordinated development of coupled systems. Especially, the digital economy keeps important support for rural logistics to make up for the shortcomings, strengthen the foundation and pursue high-quality development. And rural logistics is the connecting hub for the development of the rural digital economy.

To achieve quality development in rural areas, it is necessary to continue to improve coupling systems. The integration capacity of the digital economy is an important support for the development of rural logistics, which rationalizes the decentralized, networked, and technological model of supply and demand for agricultural materials. But it is difficult to achieve large-scale production and distribution because of the weak foundation of rural logistics. To this end, provinces need to draw on the power of the digital economy to make up for deficiency. Furthermore, the ability to integrate and innovate can facilitate digital logistics and should be taken seriously by government departments and logistics companies. With the coupling and integration between the digital economy and rural logistics, a "digital butterfly" has been created in the areas of consumption, distribution, and production. It has facilitated a leap of collaborative innovation in agriculture and rural areas, giving rise to new models and new dynamics.

## Creating a "fine, precise, and accurate" plan for quality development

In response to the development proposals of the different echelons, each province should take into account the regional differences, and formulates a high-quality development plan that is "fine, precise, and accurate". From a systemic perspective, although coordinated development should be grasped as a whole, regional differences are inevitable. It may be discussed in a differentiated manner to meet the requirements of "fine, precise, and accurate" quality development. Therefore, because of the differences in the coupled and coordinated development of the four echelons, differentiated development proposals are put forward.

On one side, the concept of regional development should be implemented to reach a new ecology of rural digital logistics. The results suggest that the variability of the CCDs across the provinces is gradually decreasing. It is evident that China is on the right track in guiding a strategic digital transformation of rural logistics. At a later stage, to break through the "bottleneck" in the development of rural logistics, the new digital logistics ecosystem needs to be cultivated and strengthened in rural areas. For this, two aspects need to be followed. First, efficiency improvements depend on two main dimensions, the improvement of infrastructure and advances in digital technology. As a key player, companies must be actively involved in digital and intelligent transformation, as well as operational efficiency and refinement. A lesson should be known for logistics companies to address the drawbacks of traditional rural logistics development. technologies such as 5G, artificial intelligence, and autonomous driving should be integrated into areas such as distribution systems. Second, rural digital and platform-based logistics systems need to be accelerated. Because the main elements influencing the development of high-quality rural logistics have been transformed into digital technology and capital input elements, etc. To make agricultural and rural economic exchanges effective, the regional industry must establish a solid digital, platform-based logistics system.

On the other side, mismatched elements have to be eliminated to balance digital and flexible logistics systems. The fourth echelon, consisting of Heilongjiang, Yunnan, Jilin, and Ningxia, is in the coupled disorder stage, showing a non-converging trend of first decreasing and then increasing. It can be seen that there are still some problems in this echelon, probably

caused by poor infrastructure, uneven network layout, low marketability, and difficulties in integrated management. In addition, scholars have also proposed that because of the large geographical differences in rural areas, the distribution range is small and the residential address is less clear, resulting in a sparse express logistics and distribution network in rural areas. In this regard, corresponding strategies need to be adopted. Combining these reasons, a scientific distribution network for digital logistics is very necessary in rural areas. Moreover, in addition to strengthening the construction of transportation facilities such as "railway and public infrastructure", geographical or human factors should be considered by the provinces. For example, the construction of logistics outlets in nodal areas with crowd-gathering characteristics, such as village committees or community hospitals, will ultimately achieve the perfection of logistics and distribution services while facilitating the lives of rural residents. Then, it is also important to develop digital talents. The training and management of digital talent needs to be extended to more areas of breakthrough. For example, training talents in material transportation, digital technology liberalization, digital technology maintenance, digital platform construction, etc.

## Supporting information

**S1 Dataset.**
(ZIP)

## Author Contributions

**Conceptualization:** Lizhen Zhan, Xiaowei Lin.

**Formal analysis:** Hui Shu.

**Investigation:** Xideng Zhou.

**Methodology:** Hui Shu, Xiaowei Lin.

**Resources:** Xiaowei Lin.

**Software:** Lizhen Zhan.

**Writing – review & editing:** Hui Shu, Lizhen Zhan, Xideng Zhou.

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
