## [Decision Letter · Decision Letter 0]

11 Oct 2022

PONE-D-22-22515Coordination Measure for Coupling System of Digital Economy and Rural Logistics:A case study of ChinaPLOS ONE

Dear Dr. Lizhen Zhan,

Thank you for submitting your manuscript to PLOS ONE. After careful consideration, we feel that it has merit but does not fully meet PLOS ONE’s publication criteria as it currently stands. Therefore, we invite you to submit a revised version of the manuscript that addresses the points raised during the review process.

We look forward to receiving your revised manuscript.

Kind regards,

Ricky Chee Jiun Chia

Academic Editor

PLOS ONE

Journal Requirements:

"Lin was funded by the National Social Science Foundation of China Western Project “Research on the dilemma, causes and strategies of high-quality development of rural logistics driven by digital economy” (No. 20XJY011).

Zhou was funded by science and Technology Research Project of Jiangxi Provincial Education Department (No. GJJ213106).

Shu was funded by the Bidding Project of Key Research Bases of Humanities and Social Sciences in Jiangxi Universities: “Research on Strategies to Promote Synergistic Development of Agricultural Logistics Ecosystem in Jiangxi Province” (JD20027)."

Reviewers' comments:

Reviewer's Responses to Questions

**Comments to the Author**

1. Is the manuscript technically sound, and do the data support the conclusions?

Reviewer #1: Yes

Reviewer #2: Partly

2. Has the statistical analysis been performed appropriately and rigorously? 

Reviewer #1: Yes

Reviewer #2: Yes

3. Have the authors made all data underlying the findings in their manuscript fully available?

Reviewer #1: Yes

Reviewer #2: Yes

4. Is the manuscript presented in an intelligible fashion and written in standard English?

Reviewer #1: No

Reviewer #2: No

5. Review Comments to the Author

Reviewer #1: 1.In the abstract, "In this paper, the primary logic relationship and operation structure of the coupling system are linked by digital economy and rural logistics, are elucidated based on system theory and coupling theory as the theoretical analysis framework." There should be an "and" between the "logistics," and "are".

2.In the abstract, " and the mean of CD is increasing year over year" . "Year over year" should be "Year by year".

3.In the 6th line of the 1st paragraph of section 6.2，“Immediately after, in the vertical analysis, the mean of CD

tends to increase year by year as the year increases.” as the year increase should be deleted.

Reviewer #2: The paper concentrates on the important issue from the perspective of regions, regional authorities and enterprises in regard to the digital economiy and rural logistics.

The subject matter of the article is interesting, but there are many areas for improvement.

The abstract lacks the purpose of the paper.

In the theoretical part, there is no reference to systems theory and coupling theory cited by the authors.

In the introduction the authors write: For example, in the logistics links such as picking, transportation and storage, it's lost by 25% to 30% for China's fruits, vegetables and other agricultural products, while the loss rate of fruits and vegetables in developed countries is below 5%. Based on what is this data (can you put reference supporting it)?

The purpose, research questions and research gap should be more marked.

I do not see sources given for the figures.

As for the conclusion, the results should be precisely contrasted with the literature described.

The evaluation analysis contains many comments that are not based on the results but on the authors' knowledge, so it would be good to move them to the discussion. You write that certain things are due to historical and geographical factors. It would be worth mentioning these in the literature review.

Methodology:

You write that 12 features were selected for the comprehensive index of coupled system, but in fact there are 11!!!.

The choice of the research period 2016-2019 is also incomprehensible to me. The authors did not justify why exactly this one and what it is due to. It seems to me that such time serious are too short and unreliable.

The table with the list of variables is missing units. A kind of robust check is also missing. Are the weights that the authors obtained significantly different from those that emerged from the TOPSIS method?

This raises the question of which of the rural logistic subsystems has the greatest impact on the digital economy? Do we know that?

I also leave it to the authors to think about the use of panel data analysis and panel cointegration, which captures not only the spatial factor but also the time factor. Using Granger causality tests, you can assess which subsystem influences each other and to what extent.

In table 5 at the bottom you write Total. I believe this is mean.

The limitations of the research should be given in the discussion or conclusions.

How do your results differ from others, what do they add?

It could be value added also to include your short thoughts regarding COVID-19 and it's impact on digital economy and rural logistics?

Editorial part.

The article is written with numerous editorial errors and some sentences are difficult to understand (e.g. One is the application of digital technologies in the form of digital technologies). Needs linguistic correction. There are many double spaces,. In chapter 6.2 you write that the research is from 2010 to 2019, while previously the research period was marked as 2016-2019. In the same sentence you indicated 0.784 to 1 but in fact I found 0.785 in your table.

References are in different formats, some articles do not indicate numbering properly. Some of the links to the electronic version of the article (doi) have been additionally numbered which is a mistake (reference 4 should be part of reference 3). Incorrect numbering of citations inside the article is also a result of this, as in the references at the end of the article everything has moved one position forward, while in the citations inside the article it has not. Incorrectly inserted commas, full stops, spaces in many places in the literature entries

6. PLOS authors have the option to publish the peer review history of their article (what does this mean?). If published, this will include your full peer review and any attached files.

Reviewer #1: No

Reviewer #2: No

---

## [Author Response · Author response to Decision Letter 0]

11 Jan 2023

Reviewer #1:

We are very grateful for your suggestions. These comments are essential for the improvement of our manuscripts. We strongly agree with your suggestions and we are keenly aware of the importance of accurate language. At the same time, we would like to say how sorry we are for the negligence that led to some grammatical errors in the manuscript. Thankfully, however, with your guidance, we have had the opportunity to correct these grammatical errors in time. In addition to revising the three obvious errors you suggested, we have also made grammatical corrections to other parts of the manuscript and marked them in red. We hope that our corrections will be accepted. If there are still areas that need to be corrected, we look forward to hearing from you and we will certainly endeavor to do our best to improve them.

Comment 1: In the abstract, “In this paper, the primary logic relationship and operation structure of the coupling system are linked by the digital economy and rural logistics, are elucidated based on system theory and coupling theory as the theoretical analysis framework.” There should be an “and” between the “logistics,” and “are”.

Response 1: We would like to express our sincere thanks to you. The sentence has been revised following your guidance. The revised text reads, “In this paper, the primary logic relationship and operation structure of the coupling system are linked by the digital economy and rural logistics, and are elucidated based on system theory and coupling theory as the theoretical analysis framework.” 

Comment 2: In the abstract, “and the mean of CD is increasing year over year”. “Year over year” should be “Year by year”.

Response 2: Thank you for the suggestion, we have corrected this sentence. Amend to read “and the mean of CD is increasing”. Moreover, we have refreshed and corrected the abstract. The corrections are as follows. “As an important engine for high-quality economic development, the digital economy is gradually integrating with the rural logistics industry. This trend is contributing to making rural logistics a fundamental, strategic, and pioneering industry. However, some valuable topics remain unstudied, such as whether they are coupled and whether there is variability in the coupling system across the provinces. Therefore, this article takes system theory and coupling theory as the analytical framework, aiming to better elaborate the subject’s logical relationship and operational structure of the coupled system, which is composed of a digital economy subsystem and a rural logistics subsystem. Furthermore, 21 provinces are seen as the research object in China, and the coupling coordination model is constructed, aiming to verify the coupling and coordination relationship between the two subsystems. The results suggest that two subsystems are coupled and coordinated in the same direction, and they feed back and influence each other. During the same period, four echelons are divided and there is variability in the coupling and coordination between the digital economy and rural logistics, according to the coupling degree (CD) and coupling coordination degree (CCD). The findings presented here can serve as a useful reference for the evolutionary laws of the coupled system. The findings presented here can serve as a useful reference for the evolutionary laws of coupled systems. And, it will further provide ideas for the development of rural logistics and digital economy.”(marked in red L18-34)

Comment 3: In the 6th line of the 1st paragraph of section 6.2, “Immediately after, in the vertical analysis, the mean of CD tends to increase year by year as the year increases.” as the year increase should be deleted.

Response 3: We apologize for the error in the statement due to an oversight. Thank you very much for your guidance. With your guidance, we have corrected it. As follows, “Immediately after, in the vertical analysis, the mean of CD tends to increase year by year.”(marked in red L477-478)

Reviewer #2:

We quite appreciate your favorite consideration and insightful comments. Now we have carefully analyzed and revised exactly according to your comments and found these comments are very helpful. We hope this revision can make our paper more acceptable. The revisions were addressed point by point below.

Comment 1: The abstract lacks the purpose of the paper. 

Response 1: Thank you for your guidance and very important suggestions. We add the purpose of the study to the abstract based on your guidance. Also, we have further modified the language description of the abstract. To highlight the revisions, we marked them in red font in the manuscript (marked in red L18-34).

Comment 2: In the theoretical part, there is no reference to systems theory and coupling theory cited by the authors. 

Response 2: Thank you very much for your valuable comments. According to your guidance, We trace the origins of two theories. Systems theory was originally established by Bertalanffy, whose monograph entitled “General System Theory; Foundations, Development, Applications”, published in 1968, is widely recognized as the masterpiece of the discipline. Subsequently, he also published an article entitled “The History and Current State of General Systems Theory”, which explored the future development of systems research and is also recognized as the basis for the development of systems theory. In addition, in 1990, Orton and Weick used coupling theory to explain socio-economic problems. Since then, coupling theory has been developed and gradually expanded to economic management and other fields, and a series of coupling degree model development results have been achieved. Therefore, we added information from the literature on systems theory and coupling theory in the manuscript and made citations(as cited in [45] and [46]).

Comment 3: Two parts need to be revised in the introduction. First, the purpose, research questions, and research gap should be more marked. Second, the authors write: For example, in the logistics links such as picking, transportation, and storage, it's lost by 25% to 30% for China's fruits, vegetables, and other agricultural products, while the loss rate of fruits and vegetables in developed countries is below 5%. Based on what is this data (can you put a reference supporting it)? 

Response 3: In accordance with your guidance, we have revised and updated the content of the introduction. To begin with, we have added references supporting the discussion in the introduction (as cited in [6]). Then, the purpose of the study, the research questions, and the research gaps were given more prominence in the introduction. These are as follows (marked in red L110-129).

Research gap: With the gradual penetration of digital elements into the real economy, digital transformation is gradually becoming a core element in the high-quality development of the rural logistics industry. Digital elements drive it to strengthen the digital foundation, make up for technical shortcomings, etc. However, current studies have mainly analyzed the importance, and impact relationships and coordination paths between the digital economy and rural logistics. Few studies have examined their logical relationship as well as regional differences from a coupling perspective. Clarifying the coupling relationship becomes an important topic.

Research questions: Specifically, this study intends to answer the following questions.(1) Based on theoretical analysis, how the coupled system consisting of the digital economy subsystem and the rural logistics subsystem operates? (2) By constructing a coupling coordination model, whether it can be verified that the digital economy subsystem and the rural logistics subsystem are both coupled? (3) Is the evolution of the coupled system set in stone? If not, are there different stages of coupling coordination, and what kind of variability is there?

Research purpose: To answer three questions, this study sets up the evaluation index and constructs a coupling and coordination model to evaluate CD and CCD. On the one hand, system theory and coupling theory are used as the analytical framework to innovatively construct the coupling system, and clarify the main logical relationship and operation structure. On the other hand, based on the analysis results of CD and CCD, the changing trends and regional differences are examined. Ultimately, suggestions are provided for the development of the coupling between the digital economy and rural logistics.

Comment 4: You write that 12 features were selected for the comprehensive index of the coupled system, but there are 11! In table 5 at the bottom you write Total. I believe this is mean. 

Response 4: We are very sorry for the error in this section due to an oversight. Fortunately, your guidance is much appreciated. On the one hand, the Evaluation index system for the rural logistics subsystem includes 2 variables, 5 secondary dimensions, and 11 evaluation indicators (e.g. A1-A2, B1-B9). On the other hand, we have amended the bottom of Table 8 by replacing the word “Total” with “Mean”.

Comment 5: As for the conclusion, the results should be precisely contrasted with the literature described. 

Response 5: We are super grateful to get your guidance. With your suggestions, we have revised the “Conclusion”, and added “Discussion”. In the discussion section, research results are accurately contrasted with the literature described(marked in red L568-601).

Comment 6: The evaluation analysis contains many comments that are not based on the results but on the authors' knowledge, so it would be good to move them to the discussion. You write that certain things are due to historical and geographical factors. It would be worth mentioning these in the literature review. 

Response 6: We can’t agree more with your views. Following your guidance, we have updated the content in the evaluation analysis(marked in red 493-510). The additional sections have been moved to the discussion(marked in red L568-601).

Comment 7: The choice of the research period 2016-2019 is also incomprehensible to me. The authors did not justify why exactly this one and what it is due to. It seems to me that such time serious is too short and unreliable. 

Response 7: Sincerely, thank you for your suggestions. Your suggestions are very relevant to our research. We have also given in-depth thought to the issue of time selection. We have considered two main aspects in the selection of data for our study. First, Yi, executive director of the China institute of modern economics, suggested that 2016 was the first year of China's digital economy, which also meant that the Chinese digital economy was moving from enlightenment to maturity. Second, we consulted statistics that confirm this view. It is only since 2016 that Chinese data on indicators of the digital economy have become more complete, enabling a better argument to be made for the relationship between the digital economy and rural logistics. In particular, we integrated data from 2016-2019 from the China Digital Economy Development Index (DEDI) Research Report (2017-2020) published by the China Electronics Information Industry Development Institute, based on their statistical measurement of the digital economy index. However, we have had concerns and we have thought deeply about whether the research period will affect the reliability of the study. We tracked down to our main research question, which was to argue for the coupling of the digital economy subsystem with the rural logistics subsystem and the variability of development across provinces. Our analysis of the data results by year and by province is a good way to argue for the coupling relationship between the interval 2016 and 2019, as well as a good research perspective. Ultimately, this paper uses the provincial data from 2016-2019 as the research sample. Nevertheless, your comments have also been very enlightening and we are acutely aware that there is much more that can be done to extend our research. In another study, we will also try to get more dimensions to our findings by broadening the data surface.

Comment 8: The table with the list of variables is missing units. 

Response 8: Thank you for your careful guidance. We are very sorry about the unit. After receiving your valuable comments, we checked them very carefully and did our best to improve and revise them. First, as our variables are dimensionless, the variables in the Digital Economy sub-system and the Rural Logistics sub-system are unitless. However, the evaluation indicators do have units, and we are very sorry that we have not added units due to our oversight. With your guidance, we have added units to the rightmost column of Table 1.

Comment 9: Are the weights that the authors obtained significantly different from those that emerged from the TOPSIS method? 

Response 9: TOPSIS is mainly used to evaluate the superiority and disadvantages of a limited number of options, by calculating the distance between the evaluation object and the optimal and inferior solutions. It is broadly applied to purchasing decisions , comparative analysis of supply chains , mission-critical planning , product selection , etc. . The TOPSIS method, gained application popularity due to its novel use of the objective weight elicitation process based on Shannon’s entropy theory . After some careful combing and reflection, we discovered that this method has some inapplicability. Therefore, we deleted TOPSIS when we revised the model. 

Reason 1: In general, TOPSIS is applied to analyze cross-sectional data to evaluate the primary and secondary ranking of the influencing factors. This article, on the other hand, is panel data and there is unsuitability in its use. 

Reason 2: The acquisition of indicator weights is triggered based on the significance of entropy. And the entropy weighting method is extended based on the TOPSIS model. This article has borrowed ideas from the TOPSIS method in the process of standardization of extreme differences. 

Reason 3：Measurement requirements of the model can be satisfied by the entropy method, and there is no need to reuse the TOPSIS method.

Nonetheless, your comments have also inspired us to make our model more reasonable. We have also gained a better insight into the TOPSIS method. We have also made further plans. In our next article, we intend to combine the model ideas of the TOPSIS method and construct a new model. It will be used to demonstrate the main factors of the digital economy subsystem and the rural logistics subsystem and to work towards a decision plan and development plan to improve the performance of the coupled system.

Comment 10: A kind of robust check is also missing. I also leave it to the authors to think about the use of panel data analysis and panel cointegration, which captures not only the spatial factor but also the time factor. Using Granger causality tests, you can assess which subsystem influences each other and to what extent. 

Response 10: Thank you very much for the suggestions, which have had a direct effect on improving the quality of the manuscript. In one respect, in Section 6, we carry out a stability test of the data. The Harris-Tzavalis unit-root test was chosen to avoid the phenomenon of “pseudo-regression” and to ensure that the results were statistically valid. The initial statement of the hypothesis was “Ho: Panels contain unit roots.” When the initial theory is disproven, it is stated as “Ha: Panels are stationary.” Furthermore, Y and X are set as the evaluation indices for the rural logistics subsystem and the digital economy subsystem respectively. And X and Y underwent the Harris-Tzavalis unit root test. Table 7 displays the results. 

Table 7. Harris-Tzavalis unit-root test for Y and X

Subsystems Variables Statistic Z P-value Test results Stabilization

Rural logistics X -0.3021 -5.5640 0.0000*** Reject Ho Yes

Digital economy Y -0.0686 -3.7133 0.0001*** Reject Ho Yes

Note: *** denotes significance at 1% confidence level

According to the results in Table 7, the test results of the evaluation indices (X and Y) of the rural logistics subsystem and the digital economy subsystem both reject the original hypothesis. Therefore, both X and Y are stable series, and we can further measure the coupling relationship. (marked in red L457-472)

In addition, according to your instructions, we also study the “Granger causality tests” carefully. The Granger causality test is a statistical method of hypothesis testing, which is based on the least squares method of testing time series . And it tests whether one set of time series x is the cause of another set of time series y (Granger, 1980) . Three scholars, Jiaqi Xiao, Yiannis Karavias, and Vasilis Sarafidis, developed the xtgranger command to make better use of the Granger causality test idea in panel data. However, the xtgranger command mainly considers the problem of Granger non-causality testing for panel data with large cross-sections and time series dimensions. Therefore, we thought further about its applicability. After some thought and discussion, we found that it is inappropriate. According to some scholars, the Granger causality test is mainly used to test the correlation between two sets of time series data. The latter extension is used for large panel data. However, the data in our paper are small panel data, so there is inapplicability. To further test this idea, we conducted Granger correlation tests for each of the 21 provinces, and only 10 provinces had significant correlations with each other. The other 11 provinces could not be tested due to insufficient data volume.

Comment 11: This raises the question of which of the rural logistic subsystems has the greatest impact on the digital economy. Do we know that?

Response 11: According to your guidance, we feel that this issue is very meaningful and we sincerely thank you for it. We have discussed this issue in depth. The discussion revealed that our study was able to know, in which specific provinces, the two subsystems are most interrelated. Therefore, we are adding to our manuscript(marked in red L476-478). The coupled system is made up of a digital economy subsystem and a rural logistics subsystem. Each subsystem is in turn a complex of behaviors influenced by several elements. A coupling coordination degree model was constructed to evaluate the coupling relationship between the rural logistics subsystem and the digital economy subsystem in each province, as well as the stage of coupling coordination they are in. From the evaluation results, it can be seen that in 2017, Liaoning province has the highest CD value, indicating that its rural logistics subsystem has the highest degree of coupling with the digital economy subsystem.

Comment 12: do not see sources given for the figures.

Response 12: We are grateful for your guidance on the sources given for the figures. We are profoundly aware that data is fundamental support for a study. With your guidance, we have added sources of figures (marked in red L326-333, L337-357). In addition, we feel it is important to clarify the acquisition of data. The details are as follows. The data in our manuscript is divided into two main parts. One part is the measured data for the rural logistics subsystem. The other part is the measured data for the digital economy subsystem. Primarily, the data for the rural logistics subsystem mainly includes 11 evaluation indicators (e.g. A1-A2, B1-B9), which are derived from data items in databases such as the China Statistical Yearbook, the China Logistics Yearbook, the China Rural Statistics Yearbook and the China Logistics Development Report. The minimum data set is shown in the Annex (Raw data for rural logistics systems). Then, the data for the digital economy subsystem mainly refer to the China Digital Economy Development Index (DEDI) Research Report (2017-2020) published by the China Electronics Information Industry Development Institute. We have also combined the information from the Blue Book on Big Data: China's Big Data Development Report (2017-2020) and the White Paper on China's Digital Economy Index (2020) to collate 21 provinces of digital economy development index data. The data we have compiled is available in the Annex (Data for the digital economy subsystem (DEDI)).

Comment 13: The limitations of the research should be given in the discussion or conclusions. 

Response 13: We strongly agree with your suggestion. We also feel that the limitations should be written in the conclusion section. Therefore, the limitations of this article include several main aspects. The system for evaluating indicators still has flaws. The qualitative dimension has not been fully accounted for by the system of indicators used to assess subsystems. Even though the selection procedure is strictly followed the paradigm. The problem of indicator bias is not taken into account and has to be calibrated and improved in various settings or provinces. Next, this study analyzes the CD and CCD in the system using panel data as a sample without considering the dynamic, ongoing effects of the coupled system. Further research on more in-depth research topics may include continual data gathering and matching, in our opinion. At last, provinces and years considered in this work have limits given the availability of data. The follow-up study has to be improved further to try to correct the sample size bias in the results(marked in red L557-566).

Comment 14: How do your results differ from others, what do they add? 

Response 14: To highlight the differences between the findings and previous studies, we have modified the conclusion and discussion. In conclusion, the results and innovation of the study have been added(marked in red L534-556). In the discussion, We describe the differences in our study compared to previous studies(marked in red L567-637).

Comment 15: It could be value added also to include your short thoughts regarding COVID-19 and it's impact on the digital economy and rural logistics. 

Response 15: Thank you very much for your guidance. Your advice is much appreciated. We feel it is valuable to think about our research in the context of COVID-19. After discussing and reflecting on the impact of COVID-19 on coupled systems, we had a great deal to learn. We think it is important to take a dialectical view. COVID-19 puts economic pressure on the digital economy and rural logistics, but at the same time, it promotes their integration and stimulates new economic development models. Then, we will briefly talk about our views on the topics, as follows.

Since the outbreak of COVID-19, the world has been affected by the decline in economic dynamism and the deterioration of the macro environment. In the context of the wider economic system, the coupled system, consisting of the digital economy subsystem and the rural logistics subsystem, will inevitably be affected. To slow down the spread of the epidemic, it is necessary to avoid gathering crowds. No doubt about it, it puts economic and social development on a “hold button”. There is a time of increased uncertainty and downward pressure on the economy. 

But epidemic prevention and control is not just a challenge. We also have to look at the important opportunities that lie within. New opportunities are got for the coupling of the digital economy and rural logistics. The spread of COVID-19 promotes the gradual “penetration” and “empowerment” of digital elements in the rural logistics industry and the development of a coupled rural logistics and digital economy. Extraordinary resilience was stimulated by the digital economy during epidemic prevention and control. The developed digital economy is the basis for the implementation of strict controls in the countryside and is also considered to be a solid foundation for a smooth transition for farmers. It not only facilitates the digital transformation of rural consumption but also builds a solid digital distribution base for the supply of agricultural production materials, farmers’ living, and consumption needs, as well as the external distribution of agricultural products. 

Coupled development is seen as an opportunity for economic transformation. Moreover, rural logistics and the digital economy need to take a “road” of stable coupling. The plight of production and living materials in some rural areas has been blocked by factors such as transportation blockades and suspension of offline transactions. This exposed some of the problems in the development of rural logistics, such as a weak industrial base, low penetration of digital technology, and weak resource integration capabilities. Thus, the urgency and importance of rural logistics development in the era of the epidemic are visible. To promote the recovery of the rural economy, the coupling development of the digital economy and rural logistics is the trend of economic development. Their coupling allows for the integration of multiple resources such as matching farmers, general contractors, distributors, logistics, and distribution to provide rural production and living materials. Therefore, while the epidemic puts pressure on the rural logistics industry, it also promotes the digital transformation of rural logistics.

Comment 16: Editorial part: The article is written with numerous editorial errors and some sentences are difficult to understand (e.g. One is the application of digital technologies in the form of digital technologies). Needs linguistic correction. There are many double spaces, In chapter 6.2 you write that the research is from 2010 to 2019, while previously the research period was marked as 2016-2019. In the same sentence, you indicated 0.784 to 1 but I found 0.785 in your table. 

Response 16: We are very sorry for our mistake. We have amended the language errors following your guidance. First, We replace “One is the application of digital technologies in the form of digital technologies” with “One is to add new impetus to rural logistics through the application of digital technology.” (marked in red L82). Second, “the research is from 2016 to 2019”, We have changed 2010 to 2016(marked in red L475). Third, We have corrected the sentence pointed out: “To begin with, from 2016 to 2019, the CD shows from 0.784 to 1, indicating that they have a coupled relationship of mutual feedback and interaction.” (marked in red L475-476)

Comment 17: References are in different formats, and some articles do not indicate numbering properly. Some of the links to the electronic version of the article (doi) have been additionally numbered which is a mistake (reference 4 should be part of reference 3). Incorrect numbering of citations inside the article is also a result of this, as in the references at the end of the article everything has moved one position forward, while in the citations inside the article it has not. Incorrectly inserted commas, full stops, spaces in many places in the literature entries. 

Response 17: Thank you very much for your suggestion. At the same time, we apologize for the oversight. With your guidance, we have re-formatted the references and text following journal requirements. We have highlighted the changes in red. We hope that the changes we have made will better meet the publication requirements.

---

## [Editor Report · Decision Letter 1]

19 Jan 2023

Coordination measure for coupling system of digital economy and rural logistics:  evidence from China

PONE-D-22-22515R1

Dear Dr. Lizhen Zhan,

We’re pleased to inform you that your manuscript has been judged scientifically suitable for publication and will be formally accepted for publication once it meets all outstanding technical requirements.

Kind regards,

Ricky Chee Jiun Chia

Academic Editor

PLOS ONE
---

## [Editor Report · Acceptance letter]

2 Feb 2023

PONE-D-22-22515R1 

Coordination measure for coupling system of digital economy and rural logistics: an evidence from China  

Dear Dr. Zhan:

I'm pleased to inform you that your manuscript has been deemed suitable for publication in PLOS ONE. Congratulations! Your manuscript is now with our production department. 

Kind regards, 

on behalf of

Dr. Ricky Chee Jiun Chia 

Academic Editor

PLOS ONE